# Composition, Distribution and Mobility Potential of the Antibiotic Resistome in Sediments from the East China Sea Revealed by Metagenomic Analysis

**DOI:** 10.3390/microorganisms13030697

**Published:** 2025-03-20

**Authors:** Xiaozhong Chen, Long Gao, Yanxue Kou, Xiaoxuan Wang, Xintong Li, Hui He, Min Wang

**Affiliations:** 1College of Marine Life Sciences, Key Laboratory of Evolution & Marine Biodiversity (Ministry of Education) and Institute of Evolution & Marine Biodiversity, Frontiers Science for Deep Ocean Multispheres and Earth System, Ocean University of China, Qingdao 266003, China; chengchuanzhi@outlook.com (X.C.); gaolong@stu.edu.ouc.cn (L.G.); kouyanxue2023@163.com (Y.K.); wangxiaoxuan5586@stu.ouc.edu.cn (X.W.); lxt7534@163.com (X.L.); mingwang@ouc.edu.cn (M.W.); 2Haide College, Ocean University of China, Qingdao 266100, China

**Keywords:** antibiotic resistance genes (ARGs), antibiotic resistome, mobile genetic elements (MGEs), human pathogen bacteria (HPB), East China Sea, metagenomic

## Abstract

Marine sediments are recognized as crucial reservoirs of antibiotic resistance genes (ARGs). However, the antibiotic resistome in sediments of the East China Sea, an area heavily impacted by human activities, has not been thoroughly studied. Here, we conducted a systematic investigation into the antibiotic resistome in these sediments using metagenomic analysis. Overall, we detected eighty ARG subtypes and nineteen ARG types. Beta-lactams were the dominant ARG type, and Gammaproteobacteria was the main ARG host in this study. Mobile genetic elements (MGEs) were not major drivers of ARG profiles. Although the ARG host communities significantly differed between the spring and autumn (*p* < 0.05), the antibiotic resistome remained stable across the two seasons. The assembly of ARGs and their hosts was governed by stochastic processes, and a high ratio of stochastic processes implied its crucial role in the assembly and stabilization of the antibiotic resistome. Co-occurrence network analysis revealed an important role of Deltaproteobacteria in the stabilization of ARG profiles across seasons. Environmental parameters (e.g., temperature and density) played certain roles in the stabilization of the antibiotic resistome between spring and autumn. Moreover, nine human pathogen bacteria (HPB) were detected in this study. We also found that the health risks caused by ARGs were relatively higher in the spring. Our results will provide a strong foundation for the development of targeted management strategies to mitigate the further dissemination and spread of ARGs in marine sediments.

## 1. Introduction

Antibiotics are antimicrobial agents that are effective against bacteria [1]. The indiscriminate use of antibiotics leads to the widespread development of antibiotic resistance genes (ARGs), which are currently deemed emerging contaminants in various environments [2]. Antibiotic-resistant microorganisms (ARMs), which harbor either an individual or multiple ARGs, have been increasingly detected in diverse environments; however, the existing antibiotics have limited or no effect on these microorganisms [3]. The severe infection and high mortality triggered by ARMs can influence humans, either through direct contact or via the food chain, posing a grave threat to contemporary health care systems [4,5].

ARGs have existed in environments for thousands of years, long before the occurrence of synthetic antibiotics [6]. The prevalent utilization of antibiotics has fueled the acceleration of ARGs dissemination and evolution, thereby enhancing their widespread occurrence in natural environments [7]. The horizontal gene transfer (HGT) of ARGs, mediated by mobile genetic elements (MGEs) such as plasmids, escalates the likelihood of diverse bacterial resistance mechanisms emerging and thereby complicates therapeutic interventions [8,9,10,11,12]. Marine sediments are rich in bacteria and serve as crucial reservoirs of ARGs [13]. Moreover, the abundance of ARGs in sediments has been demonstrated to exceed that in surrounding seawaters, emphasizing their importance in sediments [14,15]. Lu et al. conducted a large scale sampling from the Bohai Sea, Yellow Sea and the major cities along the coastline from the mouth of the Yalu River to the Yangtze River, and found that the spatial distribution of target ARGs based on the absolute abundances followed the trend of river water ≈ coastal water > the Bohai Sea > the Yellow Sea [16]. More than 300 ARGs in sediments of the Baltic Sea were identified with qPCR analysis, and these ARGs were primarily from ARG types such as aminoglycoside and beta-lactam [17]. Moreover, researchers have found 402 ARG subtypes in coastal sediments of Kuwait, with the dominant subtypes being *patA*, *adeF* and *ErmE*, which mainly originated from 34 ARG types, including beta-lactam [18].

The East China Sea, with a population of 169 million people densely distributed in coastal areas as of 2020, faces significant challenges related to antibiotic contamination [19]. The dense population increases the demand for antibiotics for both medical and agricultural purposes, contributing to higher levels of antibiotic residues in this region [20]. The large scale of aquaculture, along with the antibiotic residues carried by major rivers (e.g., the Yangtze River) also places significant pressure on this region, which contribute to antibiotic contamination [21]. Previous research has focused on the detection of several common ARGs, such as beta-lactams, at relatively small scales using qPCR analysis, as well as the influence of various environmental parameters on the antibiotic resistome; however, these studies did not provide sufficient information for a rigorous evaluation of ARG distribution patterns [14,16,22,23]. At present, metagenomics offers a novel perspective on the antibiotic resistome, and this methodology could provide a more comprehensive understanding of the composition, distribution and mobility potential of ARGs, along with their related bacterial hosts [24,25,26].

The distribution and mobility of ARGs are significantly influenced by a variety of environmental parameters, such as pH, temperature and nitrate concentration [22,27,28]. Moreover, temporal patterns, such as seasonal distribution patterns, have been observed in ARGs in diverse environments. For example, in drinking water distribution systems, seasonality significantly impacts both the abundance of ARGs and the assembly processes of the antibiotic resistome [26]. Seasonal variations in the antibiotic resistome have also been identified in the Yangtze River and the Beibu Gulf [29,30]. However, the temporal patterns in the composition, distribution and mobility potential of ARGs have rarely been explored in the East China Sea [14,16]. Relevant knowledge is essential to gain a comprehensive insight into ARGs in sediments of the East China Sea, which is vital for the management of antibiotic resistance in this region.

Herein, sediments from the East China Sea were collected from April to October 2021, and ARGs were analyzed with metagenomics in this study. We aimed to (i) reveal the temporal patterns in the composition and distribution of ARGs and their related bacterial hosts, (ii) evaluate the mobility potential of ARGs and (iii) explore the driving factors and risk assessment of ARGs in sediments of the East China Sea. This study provides a theoretical basis for the development of targeted management approaches to mitigate the pollution caused by ARGs in the East China Sea.

## 2. Materials and Methods

### 2.1. Sample Collection and Determination of Environmental Factors

A total of thirty surface sediment samples were collected from fifteen stations in the East China Sea in April (spring) to October (autumn) 2021 in this study (Appendix A). The samples obtained in April were labeled with “S”, whereas those collected in October were labeled with “A” for differentiation. After collection, all the sediment samples were placed into sterile plastic bags and frozen at −80 °C prior to nucleic acid extraction. Both surface sediment and overlying seawater were collected to measure environmental factors. Temperature and salinity were recorded in situ with a CTD (Seabird, Bellevue, WA, USA). After filtration through 0.45 μm membranes, the concentrations of ammonium (NH_4_^+^), nitrite (NO_2_^−^), nitrate (NO_3_^−^), silicate (SiO_3_^2−^) and phosphate (PO_4_^3−^) were determined in the laboratory with a QuAAtro 39 nutrient autoanalyzer (Seal Analytical, Norderstedt, Germany). The contents of total nitrogen (TN), total carbon (TC) and total organic carbon (TOC) were assessed with the Vario EL III elemental analyzer (Elementar, Hesse, Germany). The chlorophyll *a* (chl *a*) content was measured using the spectrophotometric method after acetone extraction. The results for the environmental factors are shown in Appendix A.

### 2.2. DNA Extraction and Metagenomic Sequencing Analysis

Sediment DNA was extracted with the DNeasy PowerSoil Kit (Qiagen, Hilden, GER) according to the manufacturer’s instructions. The quality of the DNA was assessed with a NanoDrop ND-2000 instrument (Thermo Fisher Scientific, Waltham, MA, USA), and its concentration was accurately quantified with a Qubit 3.0 fluorometer (Invitrogen, Waltham, MA, USA). High-quality DNA, with amounts exceeding 1 μg and concentrations greater than 10 ng·μL^−1^, was used to construct libraries with the NEBNext Ultra DNA Library Prep Kit for Illumina (NEB, Ipswich, MA, USA). The libraries were then sequenced on an Illumina NovaSeq 6000 platform using a paired-end strategy (Novogene, Beijing, China). The raw reads obtained in this study were submitted to the National Center for Biotechnology Information (NCBI) under the accession number PRJNA1160398.

### 2.3. ARG and ARG Host Analysis

The raw reads were quality-filtered with Sickl (-q 20 -l 50), the NGS QC Toolkit (-l 70 -s 20) and Fastp (-q 30) [31,32]. High-quality reads were then assembled into contigs using Megahit [33]. Contigs longer than 500 bp were used to predict the open reading frames (ORFs) with Prodigal under default settings [34]. To generate nonredundant gene sets, all the predicted genes were clustered with CD-HIT (-c 95 -aS 0.9) [35]. The relative abundance of each gene was normalized by transcripts per million (TPM) with CoverM (-p bwa-mem -m tpm). DIAMOND was employed to identify ARGs from the nonredundant gene sets against the SARG3.2 database (blastp -e 10^−6^ --ultra-sensitive), and only ARGs with an identity greater than 70% were included for downstream analysis [36]. The obtained ARGs were compared with the local NCBI nonredundant (NR) database with DIAMOND (blastp -e 10^−6^ --ultrasensitive), and the results were parsed for taxonomic information using the TaxonKit [37]. The outputs of the aforementioned analysis were regarded as ARG hosts. The ARG hosts were then manually compared with a previously constructed pathogen database to identify the ARG-carrying pathogens [38]. The relative abundance of ARG hosts and pathogens was determined on the basis of the taxonomic annotation of ARGs in each sample. Contigs annotated to ARGs were considered ARG-carrying contigs (ACCs).

### 2.4. Mobility Potential of ARGs

To assess the potential capacity of HGT between ARGs and their hosts, DIAMOND was employed to identify MGEs from all the predicted ORFs in each ACC against the NCBI NR and MobileOG databases [39]. For the NR database, MGEs were annotated and classified based on string matching with keywords such as transposase and transposon [40,41]. Meanwhile, MGEs were also categorized according to the information provided by the MobileOG database [39]. After manually removing the duplicate annotations in both databases, the relative abundance of MGEs was normalized to TPM for further analysis.

### 2.5. Statistical Analysis

The co-occurrence arrangement of ARGs and MGEs carried by the same contig was analyzed using the “gggenes” package (https://github.com/wilkox/gggenes, accessed on 17 March 2025) in R software (version 4.3.1). The temporal patterns of ARGs, ARG hosts and MGEs were examined with the “vegan” package (https://github.com/vegandevs/vegan, accessed on 17 March 2025). The relationships among ARGs, ARG hosts and MGEs coexisting with ARGs were determined using linear regression analysis and Spearman correlation analysis. The relative abundance of ARGs was visualized using a heatmap with the “funkyheatmap” package (https://github.com/funkyheatmap/funkyheatmap, accessed on 17 March 2025). Temporal patterns in the relative abundance of ARGs, MGEs and ARG hosts were shown through box plots and were tested with Student’s *t* test. Principal coordinate analysis (PCoA) was conducted to assess the similarities of ARGs and ARG hosts between the spring and autumn using the “vegan” package. To explore the assembly mechanisms of ARGs and ARG hosts, a neutral community model (NCM) and normalized stochasticity ratio (NST) were employed [42,43]. The correlation between the antibiotic resistome and environmental factors was assessed via the Mantel test. Network analysis (|ρ| > 0.6, *p* < 0.05) was utilized to evaluate the interactions between ARGs and ARG hosts, as well as among ARG hosts, and the interactions were visualized using Gephi 0.10 [44]. In addition, the health risks of ARGs were quantified based on the antibiotic resistome risk index (ARRI) [45,46].

## 3. Results

### 3.1. Composition and Distribution of ARGs

Nineteen types of ARGs were detected across all the sediment samples (Figure 1a), with a greater number observed in the spring than in the autumn (Appendix A). The dominant ARG types were beta-lactam (43.5%), rifamycin (33.0%) and multidrug (17.6%) in both the spring and autumn. The relative abundances of these ARG types were not significantly different between two seasons (*p* > 0.05) according to Student’s *t* test. A total of eighty ARG subtypes were identified in this study, with the primary subtypes being *PNGM-1* (43.5%), *Bado_rpoB_RIF* (33.2%) and *rsmA* (14.5%) (Figure 1b). Specifically, there were seventy-seven ARG subtypes in the spring, and fifty-five were detected in the autumn (Appendix A). No significant changes were noted between the spring and autumn (*R*^2^ = 0.02, *p* > 0.05, Figure 1c). Furthermore, five resistance mechanisms were observed among all the ARGs in the studied areas (Figure 1d). Antibiotic inactivation (AI, 44.3%) was the dominant mechanism, followed by antibiotic target protection (ATP, 33.7%), efflux pump (EP, 19.3%), antibiotic target alteration (ATA, 2.57%) and antibiotic target replacement (ATR, 0.02%). Overall, the dominant ARG subtypes (the top 50% in terms of relative abundance), along with their corresponding ARG types and resistance mechanisms, did not significantly differ between the spring and autumn (Figure 1e).

### 3.2. Community Composition of ARG Hosts

The ARG hosts were classified into forty-two classes, with Gammaproteobacteria (39.8%) being the dominant host, followed by Deltaproteobacteria (27.5%), Anaerolineae (8.9%), Myxococcia (6.4%) and Alphaproteobacteria (4.6%) (Figure 2a). Significant variation was observed in the ARG host communities between the spring and autumn (*R*^2^ = 0.08, *p* < 0.05, Figure 2b). For example, the relative abundance of Gammaproteobacteria decreased from 41.8% in the spring to 37.7% in the autumn, whereas Deltaproteobacteria presented a greater relative abundance in the autumn (24.8% versus 30.4%). Notable differences in the abundances of Deltaproteobacteria (*p* < 0.05) and Acidimicrobiia (*p* < 0.05) were identified across seasons (Appendix A). In addition, certain classes were discovered only in the spring or autumn. For example, Longimicrobiia and Kiritimatiellia were identified only in the spring, whereas Candidatus Muproteobacteria and Spirochaetia were unique ARG hosts in the autumn (Appendix A).

At the species level, a total of one hundred and six ARG hosts were detected, with thirty unique species in the spring and eleven in the autumn (Appendix A). *Filomicrobium* sp. (10.5%), *Haliea* sp. (7.5%) and *Photobacterium obscurum* (5.0%) presented the highest relative abundances (Figure 2c, Appendix A). Notably, nine human pathogenic bacteria (HPB), such as *Enterococcus faecium* (38.3%) and *Pseudomonas aeruginosa* (11.8%), were identified in this study, accounting for 8.4% of all ARG hosts at the species level.

### 3.3. Assessment of ARG Mobility Potential

The co-occurrence patterns of ARGs and MGEs were investigated to evaluate the mobility potential of ARGs. Transposase (37.9%) was the major MGE in this study. Other MGEs, such as replication/recombination/repair (25.0%), transfer (16.0%), recombinase/integrase (8.7%), phage (8.6%) and recombinase/integrase (3.8%), were also detected to coexist with ARGs (Figure 3a). The relative abundance of MGEs decreased slightly in the autumn compared with the spring; however, the variation was not statistically significant. Notably, no significant linear correlation was identified between the relative abundances of ARGs and their coexisting MGEs (*r* = 0.08, *p* > 0.05, Appendix A), suggesting that MGEs might not be major drivers of the antibiotic resistome.

Most ACCs had only a single ARG co-occurring with one MGE, whereas some ARGs co-occurred with multiple MGEs (Figure 3b, Appendix A). The ACCs with ARGs and MGEs were primarily observed in Gammaproteobacteria, which presented diverse co-occurrence patterns between ARGs and MGEs (Figure 3b, Appendix A). Other vectors included Deltaproteobacteria, Anaerolineaceae, Arenicellales, *Candidatus* Venteria ishoeyi, *Enterococcus faecium* and *Desulforhopalus vacuolatus*, implying the occurrence of HGT among these hosts. Notably, *Enterococcus faecium* belonging to HPB was observed to harbor an ACC of three MGEs coexisting with an ARG (Figure 3b), which indicated a greater risk of *Enterococcus faecium* in ARG transfer. *rsmA* was the dominant ARG co-occurring with MGEs (Figure 3c), which suggested that it had the greatest potential for HGT in this study. Other ARGs, such as *arnA*, *tetL*, *PNGM-1*, *MexF*, *OprN* and *B_rpoB_RIF*, were also found to coexist with MGEs (Figure 3c), indicating their potential participation in HGT in the studied sediments.

### 3.4. Relationships Among ARGs, MGEs and ARG Hosts

According to Procrustes analysis, ARGs and ARG hosts were consistently dispersed, whereas ARGs and MGEs were uniformly distributed. These results suggested that ARGs had a stronger connection with ARG hosts (M^2^ = 0.7519) than with MGEs (M^2^ = 0.8585, Figure 4). Linear fitting analysis also revealed a similar trend, with a stronger connection noted between ARGs and their hosts (*r* = 0.96, *p* < 0.001, Appendix A) than MGEs (*r* = 0.08, *p* > 0.05, Appendix A), indicating a considerably lower risk caused by reduced migration rates of HGT along the East China Sea.

Network analysis revealed remarkable variations in the co-occurrence of ARGs and ARG hosts between two seasons (Figure 5). The network in the spring contained more nodes and edges (81 and 117) than those in the autumn (80 and 90), suggesting that the ARG hosts carried a greater number of ARGs and had more complex associations with them in the spring. The percentage of positive links between ARGs and their hosts increased from 70.09% in the spring to 80.81% in the autumn, indicating a stronger synergistic effect between ARGs and their hosts in the autumn. The key nodes in the network also varied across seasons. In the spring, the key nodes included Deltaproteobacteria, Deferribacteres, Halobacteria, *LlmA* and *arlR* (Figure 5a). Conversely, most edges in the network in the autumn were dominated by Deltaproteobacteria, Candidatus Dadabacteria, Epsilonproteobacteria, Flavobacteriia and *Bado_rpoB_RIF* (Figure 5b). Some crucial ARG hosts had distinct roles in the co-occurrence pattern across seasons. For example, Deltaproteobacteria manifested primarily negative correlations with ARGs in the spring network, but positive associations in the autumn network.

### 3.5. Driving Forces and Risk Assessment of ARGs

NCM was applied to quantify the assembly processes of ARGs and their hosts in this study. High interpretation rates between relative abundance and frequency occurrence were observed for both ARGs (*R*^2^ = 0.729, Appendix A) and their hosts (*R*^2^ = 0.725, Appendix A), indicating a major role of stochastic processes in the assembly of ARGs and their hosts in the studied areas. Moreover, the assembly of ARGs was less strongly explained by stochastic processes in spring (*R*^2^ = 0.594, Figure 6b) than in autumn (*R*^2^ = 0.668, Figure 6a), as was the case for ARG hosts (*R*^2^ values of 0.584 in spring and 0.679 in autumn, Figure 6a,b). These results suggested that stochastic processes might be more important in shaping ARGs and their hosts in the autumn. The normalized stochastic ratio (NST) further supported these results. The NST values of ARGs (56.2% in the spring and 68.1% in the autumn) and ARG hosts (56.0% in the spring and 69.2% in the autumn) exceeded 50% in both seasons (Figure 6c), implying that stochastic processes, rather than deterministic processes, potentially had a greater effect on the assembly of ARGs and ARG hosts in the present study.

The relationships between environmental factors and ARG profiles were examined using a Mantel test (Figure 7a). In the spring, significant correlations were observed between ARG profiles and both temperature (*r* = 0.19, *p* < 0.05) and density (*r* = 0.21, *p* < 0.05). Otherwise, salinity and NO_2_^−^ concentration were also associated with ARGs, although these correlations were not statistically significant (*p* > 0.05). Temperature, density, salinity and TC content had impacts on the antibiotic resistome in the autumn, but these effects were not significant (*p* > 0.05). The differences in the correlations between ARGs and environmental factors (e.g., temperature and density) across seasons potentially suggested that these environmental factors play a certain role in regulating the seasonal stability of ARGs in this study.

Only a small fraction (7.5%) of the top 50% of ARG subtypes belonged to Rank I, indicating that high-risk ARGs were not prevalent in the studied areas (Figure 1e). In this study, the environmental risk posed by ARGs was also evaluated with the ARRI. The average value of the ARRI was greater in the spring (1.2) than in the autumn (0.7), probably due to the greater relative abundance of HPB and ARGs co-occurring with MGEs in the spring (Figure 7b).

## 4. Discussion

### 4.1. Distribution Patterns of ARGs and ARG Hosts in Sediments of the East China Sea

There were eighty ARG subtypes and nineteen ARG types in this study (Figure 1). Beta-lactam antibiotic genes (BLRGs) were the dominant ARGs, which was consistent with previous research on the East China Sea using qPCR analysis [14]. Numerous studies have indicated the extensive distribution of BLRGs, such as penicillins and cephalosporins, in the coastal waters of China, which might be partially attributed to their extensive use, especially in clinical treatment [19,28,29,47,48]. Rifamycin resistance genes (RAGs) were also among the most dominant ARGs identified in this study (Figure 1a). Despite not being widely used, high abundance of rifamycin has been reported in diverse habitats, which could be attributed to its natural synthesis by bacteria and the presence of corresponding ARGs in different environments [48,49,50]. For example, RAGs were highly abundant in both the water and sediment of Lake Alalay in Bolivia [51]. Similarly, a high abundance of RAGs was also detected in Bohai Bay and its surrounding rivers, including the Haihe River, Yongdingxin River and Luanhe River [48].

Notably, some of the dominant ARGs identified in this study differed from those reported in the coastal areas of this region using qPCR analysis. Chen et al. (2019) noted that sulfonamide resistance genes (SRGs) were the most prevalent ARGs in the coastal sediments of this region. However, SRGs were rarely found in our study (Figure 1a), presumably because of their decreasing abundance with increasing distance from the shore, as well as microbial degradation in natural environments [48,52]. Vancomycin antibiotic genes (VRGs) were not detected in this study, contrary to the findings of other studies [53]. The abundance of VRGs varied significantly across habitats. For example, high levels were detected in human feces, wastewater treatment plant influent and animal farm manure and wastewater, whereas lower or even negligible levels were detected in river water, drinking water and sediment. These results suggested that VRGs might spread under selective pressure from the use of vancomycin, which still warrants further attention [54].

Multidrug resistance genes (MDRGs), which are part of the transporter system superfamily and ancient genes found in microorganisms, play crucial roles in facilitating the transport of various substances across cell membranes [55,56]. The presence of MDRGs promotes the emergence and dissemination of multidrug-resistant bacteria that harbor numerous ARGs, allowing these bacteria to tolerate different antibiotics and thrive in various environments [57]. In this study, MDRGs constituted 17.6% of the total ARGs, ranking third among all ARG types (Figure 1a). The principal mechanism by which these bacteria acquire resistance involves an efflux pump that expels drugs from the cell membrane or periplasmic space into the extracellular space [58]. This also explained the greater relative abundance of efflux pumps observed in this study (Figure 1d). The prevalence of MDRGs is currently of greater concern; however, there is no effective strategy to directly disrupt their emergence and spread at the source. Although efflux pump inhibitors have the potential to limit antibiotic discharge and combat MDRGs, such treatments have not been widely adopted in clinical practice.

Proteobacteria have been identified as the primary ARG hosts in marine sediments in previous studies, and a similar result was also observed in our study [14,48,59]. Among the Proteobacteria, Gammaproteobacteria were the dominant ARG hosts at the class level, both in the spring and autumn (Figure 2a,c). According to prior research, Gammaproteobacteria were considered vital taxa that respond to pollutants, such as oxytetracycline, as well as the core taxa in the gut, which was closely related to the abundance of ARGs [60]. In this study, we observed three contigs affiliated with Gammaproteobacteria that contained both ARGs and MGEs, thereby promoting the expansion of ARGs with high HGT potential due to the presence of MGEs (Figure 3b, Appendix A). Deltaproteobacteria were the second major ARG host within Proteobacteria, and their relative abundance exhibited considerable variations between the spring and autumn (Figure 2a, Appendix A). Variations in temperature directly impacted the relative abundance of Deltaproteobacteria and led to variations in ARG profiles, potentially explaining the strong correlation between temperature and ARG profiles in this study (Figure 7a) [61]. Moreover, the effects of Deltaproteobacteria on ARG profiles also presented seasonality. The network analysis revealed that Deltaproteobacteria had a primarily negative influence on ARGs in the spring, but a positive effect in the autumn (Figure 5a,b). A reduction in the relative abundance of most ARG hosts was noted in the autumn compared with the spring. However, Deltaproteobacteria exhibited the opposite trend, which might be an important factor in maintaining the balance of the antibiotic resistome across the two seasons in this study.

### 4.2. Mobility Potential and Health Risk of ARGs in Sediments of the East China Sea

MGEs, such as transposases, carry ARGs and facilitate HGT to increase the spread of bacterial resistance [62]. Our results indicated that transposase was the main MGE in the studied sediments, which was consistent with previous findings that transposase was more abundant in antibiotic-contaminated river sediments and industrially polluted lakes [63,64]. Other MGEs, including replication/recombination/repair, transfer, recombinase/integrase and phage, were also found to coexist with ARGs (Figure 3a). ARG mobility is determined by the number of their related MGEs [65]. High correlations between ARGs and MGEs have been reported in deep ocean sediments and aquaculture ponds [24,66]. In this study, most ACCs carrying MGEs had only one ARG coexisting with one MGE, but some ARGs co-occurred with multiple MGEs (Figure 3b, Appendix A). MGEs played a more pronounced role in ARG profiles in the surface water of the Beibu Gulf in the winter than in the summer [29]. However, the types and relative abundance of MGEs did not exhibit significant seasonal variations in this study. In addition, their impact on ARGs was not as strong as that of ARG hosts, as illustrated by the Procrustes and Spearman correlation analyses (Figure 4b, Appendix A). These results suggested that HGT among ARGs might not be particularly active in sediments of the East China Sea and that a lower relative abundance of MGEs might limit the further spread of ARGs, which could have contributed to the stabilization of ARG profiles across seasons in this study.

In this study, a contig belonging to *Enterococcus faecium* that poses a significant health risk was identified. *E. faecium*, colonizing humans and animals, is strongly antibiotic-resistant and persists in clinical settings. The extensive use of antibiotics in hospitals has facilitated the emergence of vancomycin-resistant *E. faecium*, a major contributor to many hospital-acquired infections [67]. In this study, the *tetL* gene, an ARG subtype associated with resistance to tetracycline, was found to be integrated within *E. faecium*. Additionally, three simultaneous MGEs were located within 1500 bp of *tetL*, indicating the high potential of HGT derived from other bacterial genomes [68]. Moreover, eight additional HPB strains were also discovered to harbor ARGs. Although there are relatively fewer HPB strains harboring ARGs in the East China Sea than in the Yangtze River, the pathogenic risk associated with ARGs should not be underestimated [30]. Otherwise, the average ARRI was higher in the spring than in the autumn (Figure 7b), likely because of the greater relative abundance of HPB and ARGs co-occurring with MGEs in the spring. This indicates more active HGT in the spring, suggesting more attention to the antibiotic resistome in the spring. However, the ARRI risk assessment, similar to many ARG risk assessment methods, primarily relies on ARG abundance, which may not accurately reflect the actual resistance risk and potential harm to humans. Consequently, there is an urgent need to establish a more scientific and systematic evaluation model to comprehensively assess the resistance risk in marine sediments.

### 4.3. Factors That Regulate the Antibiotic Resistome in Sediments of the East China Sea

The antibiotic resistome was closely related to the shift in the community of ARG hosts, indicating that ARG hosts might be a potential biotic factor that impacts ARGs in this study. The existence of a notable linear correlation between ARGs, MGEs and HPB has been documented in previous research, primarily attributed to the greater propensity of pathogenic bacteria to acquire ARGs compared with their nonpathogenic counterparts under same situation [38,48,69]. However, we observed that the relationships among ARGs, MGEs and HPB were not significant, probably because the assembly processes of ARGs were governed more strongly by stochastic processes (Figure 6, Appendix A). In addition to biotic factors, variations in ARGs are also correlated with other environmental factors, such as antibiotic concentrations and heavy metal contents [70,71,72]. Anthropogenic contamination can also influence the abundance and distribution of ARGs. Guo et al. demonstrated that heavy metals and antibiotic residues, such as oxytetracycline, doxycycline hydrochloride and norfloxacin, were significantly positively correlated with ARG abundance in sediments of the Yangtze River Estuary, indicating that both heavy metals and antibiotic residues play a critical role in promoting the spread of ARGs in this region [73]. Additionally, the discharge of domestic sewage was the primary source of ARGs in Daya Bay, with the dominant ARG types being sulfonamide and chloramphenicol resistance genes (e.g., *sul1*, *floR* and *cmlA*), which implied that anthropogenic pollution contributes to the spread of ARGs [74]. In this study, ARG hosts were correlated with season (Figure 2b). However, the effects of season on ARGs seemed to lag behind those of ARG hosts, suggesting that season might affect ARGs indirectly by altering the community of ARG hosts. Interestingly, the composition antibiotic resistome remained stable across seasons (Figure 1c and Figure 2b). NCM and NST analyses revealed that the assembly of ARGs and their hosts was governed by stochastic processes. A greater ratio of stochastic processes affecting ARGs was observed in the autumn than in the spring, indicating that stochastic processes were critical in the stabilization of the antibiotic resistome across seasons (Figure 6, Appendix A). Deltaproteobacteria presented a relatively high relative abundance (24.8% in the spring and 30.4% in the autumn, Figure 2a) and had a relatively positive influence on the antibiotic resistome in autumn (Figure 5a,b), implying its potential importance in maintaining the stability of the antibiotic resistome across seasons. Moreover, environmental factors (e.g., temperature and density) had different effects on the antibiotic resistome in the autumn compared with the spring, suggesting that these factors potentially play certain roles in the stabilization of the antibiotic resistome between the spring and autumn (Figure 7a).

## 5. Conclusions

The composition, distribution and mobility potential of the antibiotic resistome in sediments of the East China Sea were systematically investigated through metagenomics in this study. No significant differences in the antibiotic resistome were noted between the spring and autumn, but differences were observed in ARG hosts. Stochastic processes, Deltaproteobacteria and environmental factors such as temperature significantly contributed to the stabilization of the antibiotic resistome across seasons. In addition, the health risks caused by ARGs were lower in the autumn compared with the spring, suggesting that ARGs and MGEs could pose health risks through their integration into HPB, despite most ARG hosts not being pathogenic. ARGs can affect humans via food chains or other pathways; hence, investigating their transfer mechanisms along with potential pathogenic bacterial hosts could provide valuable data for public health risk assessment.

## Figures and Tables

**Figure 1 microorganisms-13-00697-f001:**
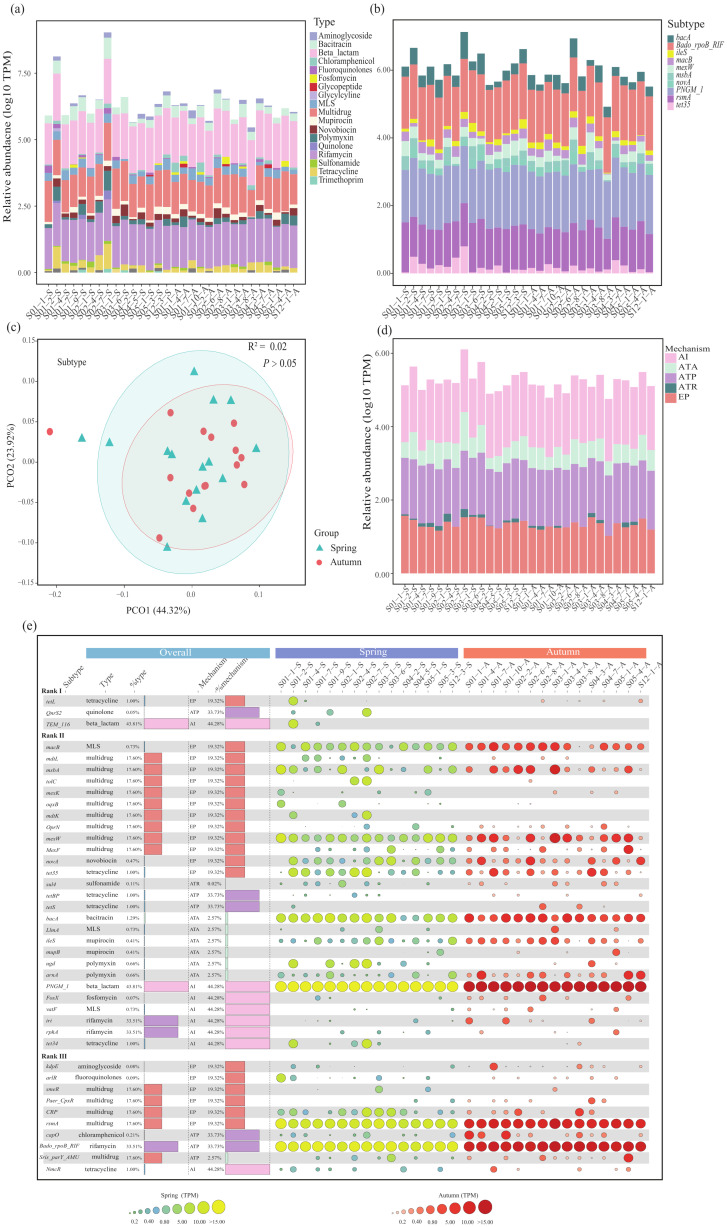
Composition of ARG profiles. (**a**) The ARG types identified in this study and their relative abundance. (**b**) The top ten ARG subtypes and their relative abundance. (**c**) PCoA analysis of ARG subtypes. (**d**) The resistance mechanisms detected in this study. EP: efflux pump, AI: antibiotic inactivation, ATR: antibiotic target replacement, ATP: antibiotic target protection, ATA: antibiotic target alteration. (**e**) The top 50% of ARG subtypes in relative abundance, along with their associated ARG types and resistance mechanisms. The type and % type respectively represent the associated ARG type of ARG subtype and its relative abundance. The mechanism and % mechanism respectively represent the corresponding mechanism of ARG subtype and its relative abundance. Rank I, II and III represent the risk level of ARG as rated in the SARG3.2 database.

**Figure 2 microorganisms-13-00697-f002:**
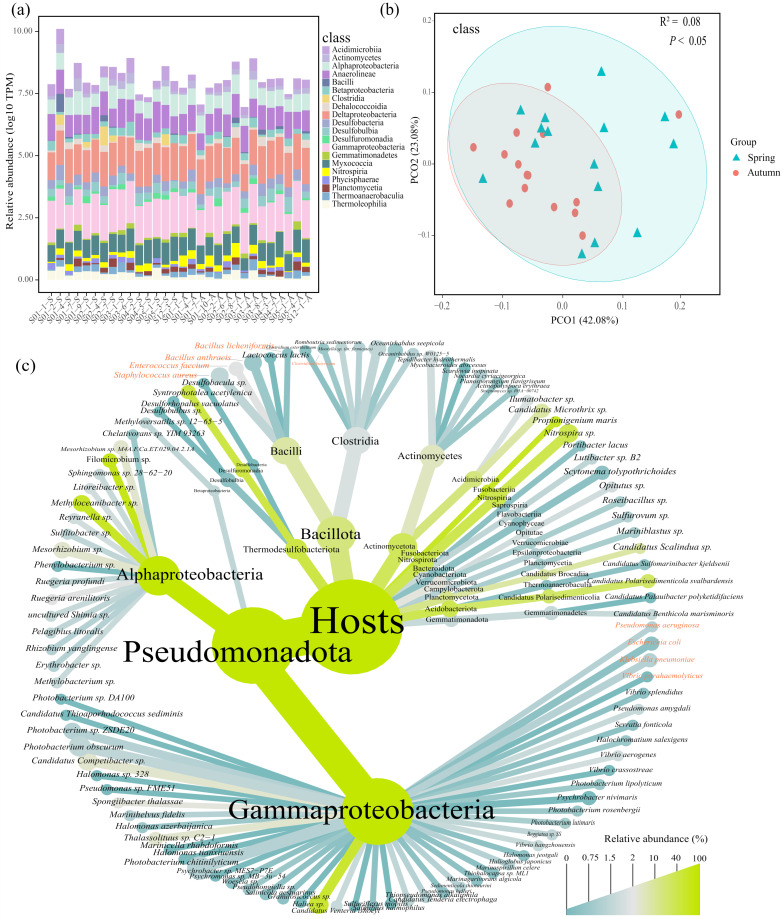
Composition of ARG hosts. (**a**) The top 50% of ARG hosts at the class level identified in this study. (**b**) PCoA analysis of ARG hosts at the class level. (**c**) Tree diagram of ARG hosts at the species level. The root represents all hosts annotated at the species level, and the nodes display their relative abundance at the phylum, order and species levels. HPB is highlighted in red.

**Figure 3 microorganisms-13-00697-f003:**
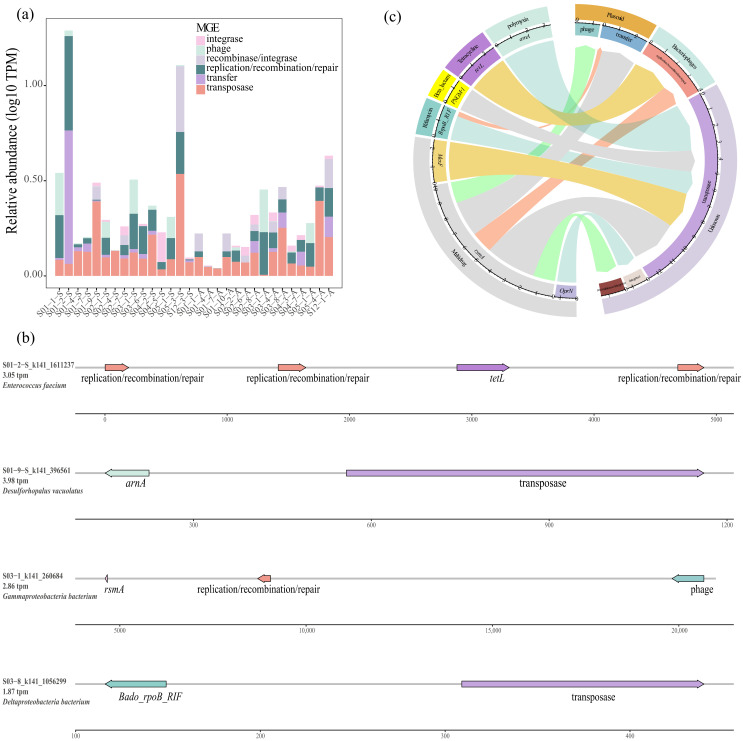
Co-existence and composition of MGEs and ACCs. (**a**) The relative abundance of MGEs. (**b**) The co-existence arrangements among ARGs and MGEs at the contig level. (**c**) The composition of ARGs transferred by MGE in sediments of the East China Sea.

**Figure 4 microorganisms-13-00697-f004:**
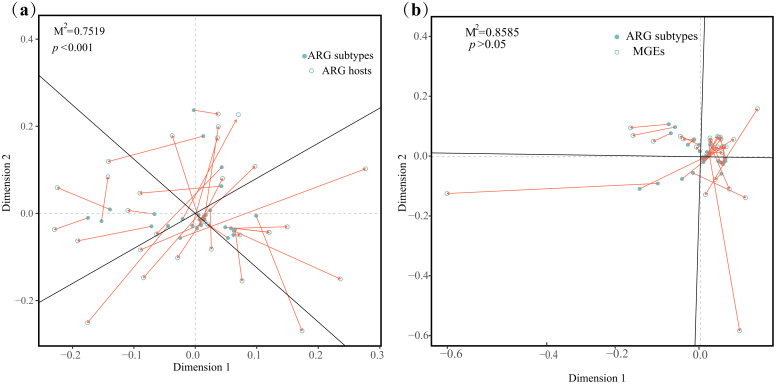
Procrustes analysis on correlations of ARG subtypes with ARG hosts (**a**) and MGEs (**b**).

**Figure 5 microorganisms-13-00697-f005:**
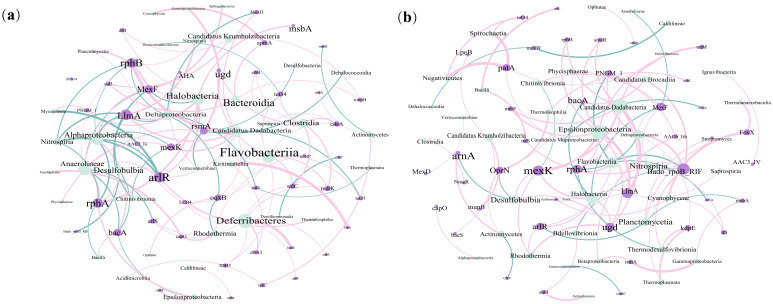
Network analysis of ARGs and ARG hosts in spring (**a**) and autumn (**b**) in sediments of the East China Sea. Size of each node corresponded to the degree within the network. Positive and negative correlations were shown as pink and blue lines, respectively.

**Figure 6 microorganisms-13-00697-f006:**
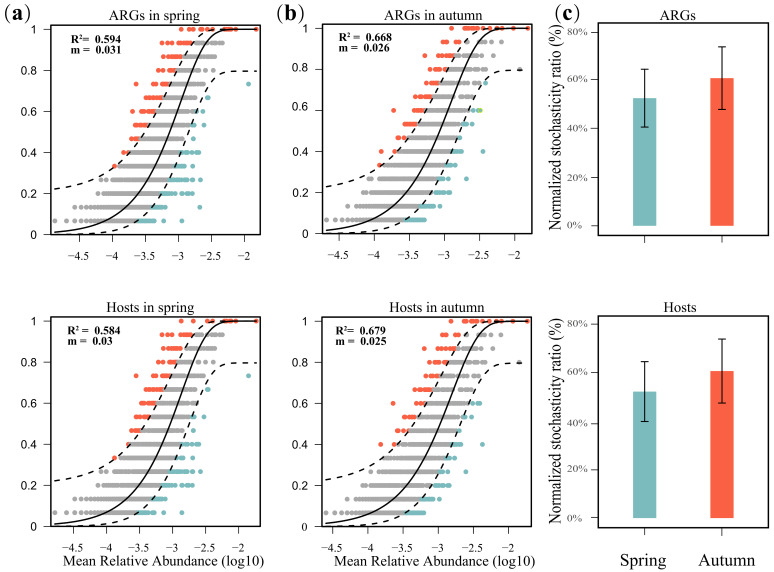
(**a**) The NCM analysis of ARGs and their hosts in the spring. (**b**) The NCM analysis of ARGs and their hosts in the autumn. (**c**) The NST analysis of ARGs and their hosts.

**Figure 7 microorganisms-13-00697-f007:**
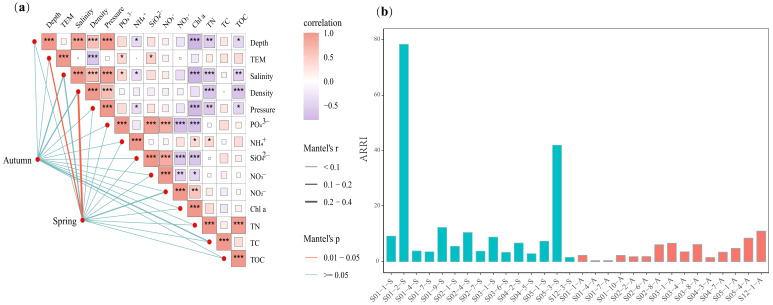
Environmental drivers and health risks of ARG profiles. (**a**) Mantel test of environmental factors and the relative abundances of ARGs. (**b**) ARRI analysis of ARGs in sediments of the East China Sea. *** *p* ≤ 0.001, ** *p* ≤ 0.01, * *p* ≤ 0.05.

## Data Availability

The raw reads were submitted to National Centre for Biotechnology Information (NCBI) under the accession number PRJNA1160398. (https://www.ncbi.nlm.nih.gov/bioproject/PRJNA1160398/, accessed on 25 February 2025).

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
