# Peer review of "Composition, Distribution and Mobility Potential of the Antibiotic Resistome in Sediments from the East China Sea Revealed by Metagenomic Analysis"

_microorganisms, 2025, doi:10.3390/microorganisms13030697_

Round 1
Reviewer 1 Report
Comments and Suggestions for Authors
After some revisions are performed, this manuscript can be considered for publication in Microorganisms.
I suggest the authors prepare a structured abstract. The study’s objectives and applied methodologies have to be clarified. Future perspectives should be indicated.
In the Introduction, the authors should also give the background of the addressed topics from a worldwide perspective. In its current form, it too much focused in the East China Sea.
The mobility potential of ARGs can be better explained (section 2.4).
The quality of all figures must be improved and their size increased. In its current state, it is very hard to understand.
The Discussion could be divided into subsections, aligned with the Results. This would facilitate the reading and understanding of the developed work.
The study’s limitations and strengths should be indicated and discussed.
The Conclusions are fine.
Author Response
Comments 1: I suggest the authors prepare a structured abstract. The study’s objectives and applied methodologies have to be clarified. Future perspectives should be indicated.
Response 1: Thank you for this comment and we accepted this comment. The objectives and applied methodologies “However, the antibiotic resistome in sediments of the East China Sea, an area heavily impacted by human activities, has not been thoroughly studied. Here, we conducted a systematic investigation into the antibiotic resistome in these sediments using metagenomic analysis.”, as well as the future perspectives “Our results will provide a strong foundation for the development of targeted management strategies to mitigate the further dissemination and spread of ARGs in marine sediments.” have been added in Abstract part in revised manuscript.
Comments 2: In the Introduction, the authors should also give the background of the addressed topics from a worldwide perspective. In its current form, it too much focused in the East China Sea.
Response 2: Thank you for this comment and we accepted this comment. “Lu et al. conducted a large scale sampling from Bohai Sea, Yellow Sea and the major cities along the coastline from the mouth of Yalu River to the Yangtze River, and found that the spatial distribution of target ARGs based on the absolute abundances was in the trend of river water ≈ coastal water > the Bohai Sea > the Yellow Sea [16]. More than 300 ARGs in sediments of the Baltic Sea with identified with qPCR analysis, and these ARGs were primarily from ARG types such as aminoglycoside and beta-lactam [17]. Moreover, researchers have found 402 ARG subtypes in coastal sediments of Kuwait, with the dominant subtypes being patA, adeF, and ErmE, which were mainly originated from 34 ARG types, including beta-lactam [18].” has been added in Introduction part.
Comments 3: The mobility potential of ARGs can be better explained (section 2.4).
Response 3: Thank you for this comment and we accepted this comment. The classification and annotation methods of MGEs “For the NR database, MGEs were annotated and classified based on string matching with keywords such as transposase and transposon [40,41]. Meanwhile, MGEs were also categorized according to the information provided by the MobileOG database [39]. After manually removing the duplicate annotations in both databases, the relative abundance of MGEs was normalized to TPM for further analysis.” has been added in Section 2.4.
Comments 4: The quality of all figures must be improved and their size increased. In its current state, it is very hard to understand.
Response 4: Thank you for this comment. We accepted this comment, and have carefully enhanced all figures in this manuscript to improve their clarity and make them more understandable.
Comments 5: The Discussion could be divided into subsections, aligned with the Results. This would facilitate the reading and understanding of the developed work.
Response 5: Thank you for this comment. We accepted this comment, and the Discussion part has been divided into subsections in revised manuscript.
Comments 6: The study’s limitations and strengths should be indicated and discussed. The Conclusions are fine.
Response 6: Thank you for this comment. We accepted this comment, and the strengths “The composition, distribution and mobility potential of antibiotic resistome in sediments of the East China Sea were systematically investigated through metagenomics in this study.” has been added in Conclusion part in revised manuscript. And the limitations “However, the ARRI risk assessment, similar to many ARG risk assessment methods, primarily relies on ARG abundance, which may not accurately reflect the actual resistance risk and potential harm to humans. Consequently, there is an urgent need to establish a more scientific and systematic evaluation model to comprehensively assess the resistance risk in marine sediments.” have been added in Discussion part in revised manuscript.
Reviewer 2 Report
Comments and Suggestions for Authors
The study provides valuable insights into the antibiotic resistome in East China Sea sediments using metagenomic analysis. The findings on the seasonal stability of antibiotic resistance genes and their hosts contribute to a better understanding of microbial resistance in marine environments. The methodology is robust and well-documented, and the statistical analyses support the conclusions effectively.
There are a few areas that could be improved. The language, while generally clear, could be refined for better readability. Some sentences are complex and could be simplified to enhance clarity. There are minor grammatical inconsistencies that should be addressed. The figures and tables are well-organized but could benefit from clearer labeling and more detailed explanations in the figure legends to ensure easy interpretation by readers.
The discussion section is comprehensive and contextualizes the findings well. However, some points could be expanded, particularly regarding the implications of horizontal gene transfer and its potential risks in marine ecosystems. Additionally, more emphasis on how these findings could inform policy and management strategies for antibiotic resistance would strengthen the impact of the study.
The study is well-executed and makes a significant contribution to the field. Addressing these minor improvements would enhance its clarity and impact.
The quality of the images can be improved (resolution).
A map with the sampling locations can be included, and the influence of anthropogenic contamination (effluent, rivers, cities, antimicrobials, antibiotics) can be discussed.
Comments on the Quality of English LanguageThe quality of English is generally good, but there is room for improvement in clarity and readability. Some sentences are overly complex and could be simplified for better understanding. There are minor grammatical inconsistencies and awkward phrasing in some sections that should be revised. Proofreading by a native English speaker or a professional editor would enhance the overall readability and flow of the text. Improving sentence structure and word choice would also help ensure that the research findings are communicated more effectively to a broader audience
Author Response
Comments 1: There are a few areas that could be improved. The language, while generally clear, could be refined for better readability. Some sentences are complex and could be simplified to enhance clarity. There are minor grammatical inconsistencies that should be addressed. The figures and tables are well-organized but could benefit from clearer labeling and more detailed explanations in the figure legends to ensure easy interpretation by readers.
Response 1: Thank you for this comment. We accepted this comment, and have simplified some sentences, “The principal mechanism by which these bacteria acquire resistance involves an efflux pump that expels drugs from the cell membrane or periplasmic space into the extra-cellular space”, “According to prior research, Gammaproteobacteria were considered vital taxa that respond to pollutants such as oxytetracycline, as well as the core taxa in the gut, which was closely related to the abundance of ARGs” and “E. faecium, colonizing humans and animals, is strongly antibiotic resistant and persists in clinical settings.” has been added in Dicussion. We also formatted all figures and enlarged the fonts within them to ensure easier interpretation.
Comments 2: The discussion section is comprehensive and contextualizes the findings well. However, some points could be expanded, particularly regarding the implications of horizontal gene transfer and its potential risks in marine ecosystems. Additionally, more emphasis on how these findings could inform policy and management strategies for antibiotic resistance would strengthen the impact of the study.
Response 2: Thank you for this comment, and we accepted this comment. “Otherwise, the average ARRI was higher in spring than autumn (Fig. 7b), likely because of the greater relative abundance of HPB and ARGs co - occurring with MGEs in spring. This indicates more active HGT in spring, suggesting more attention to the antibiotic resistome in spring.” was added in Conclusion part in revised manuscript. “In addition, the health risks caused by ARGs were lower in autumn compared with spring, suggesting that ARGs and MGEs could pose health risks through their integration into HPB despite most ARG hosts not being pathogenic. ARGs can affect humans via food chains or other pathways, hence, investigating their transfer mechanisms along with potential pathogenic bacterial hosts could provide valuable data for public health risk assessment.” was added in Conclusion part in revised manuscript.
Comments 3: The study is well-executed and makes a significant contribution to the field. Addressing these minor improvements would enhance its clarity and impact. The quality of the images can be improved (resolution).
Response 3: Thank you for this comment. We accepted this comment, and have carefully enhanced all figures in this manuscript to improve their quality.
Comments 4: A map with the sampling locations can be included, and the influence of anthropogenic contamination (effluent, rivers, cities, antimicrobials, antibiotics) can be discussed.
Response 4: Thank you for this comment, and we accepted this comment. The map of sampling locations was illustrated in Appendix A Figure S1, and detailed information of each station was presented in Appendix B Table S1. Discussion on anthropogenic pollution “Anthropogenic contamination can also influence the abundance and distribution of ARGs. Guo et al. demonstrated that heavy metals and antibiotic residues, such as oxytetracycline, doxycycline hydrochloride and norfloxacin, were significantly positively correlated with ARG abundance in sediments of the Yangtze River Estuary, indicating that both heavy metals and antibiotic residues play a critical role in promoting the spread of ARGs in this region [73]. Additionally, the discharge of domestic sewage was the primary source of ARGs in Daya Bay, with the dominant ARG types being sulfonamide and chloramphenicol resistance genes (e.g., sul1, floR and cmlA), which implied that anthropogenic pollution contributes to the spread of ARGs [74].” has been added in Discussion part in revised manuscript.
Reviewer 3 Report
Comments and Suggestions for Authors
Below are my suggestions:
A summary table comparing the seasonal variation of dominant ARG types and their hosts would improve clarity.
The text frequently repeats information that is already presented in tables and figures. Instead of restating values, focus on interpreting their significance.
Author Response
Comments 1: A summary table comparing the seasonal variation of dominant ARG types and their hosts would improve clarity.
Response 1: Thank you for this comment. The relative abundance of dominant ARG types and their hosts have been documented in Appendix B Tables 2 to 7, and sorted them based on their abundance. However, since a particular ARG may have multiple hosts, sometimes may be several or even dozens of hosts that contain a same ARG, so we did not place ARGs and their corresponding hosts in a same table.
Comments 2: The text frequently repeats information that is already presented in tables and figures. Instead of restating values, focus on interpreting their significance.
Response 2: Thank you for this comment. We accepted this comment and made an effort to minimize the repeat information already presented in figures and tables. We have tried our best to ensure the relevant information remain concise in revised manuscript.